# Identification and Characterization of Sex-Biased miRNAs in the Golden Pompano (*Trachinotus blochii*)

**DOI:** 10.3390/ani12233342

**Published:** 2022-11-29

**Authors:** Liping Shi, Feibiao Song, Kaixi Zhang, Yue Gu, Jinghan Hu, Junlong Sun, Zhongwei Wang, Li Zhou, Jian Luo

**Affiliations:** 1State Key Laboratory of Marine Resource Utilization in South China Sea, Hainan Aquaculture Breeding Engineering Research Center, Hainan Academician Team Innovation Center, Hainan University, Haikou 570228, China; 2State Key Laboratory of Freshwater Ecology and Biotechnology, Institute of Hydrobiology, Chinese Academy of Sciences, Wuhan 430072, China

**Keywords:** *Trachinotus blochii*, miRNA, gonad, sex differentiation

## Abstract

**Simple Summary:**

We performed transcriptome sequencing of the ovaries and testis of *T. blochii* to acquire sex-biased miRNAs. RNA-seq analysis yielded 1366 known and 69 novel miRNAs with 289 significant DEMs (*p* < 0.05). KEGG analysis indicated that the target genes of the DEMs were mainly enriched in pathways connected to sex differentiation and gonadal development signals, consisting of the MAPK signaling pathway, Wnt signaling pathways, and steroid biosynthetic pathways. Co-expression network analysis of miRNA-mRNA indicated that some miRNAs (oar-let-7b, bta-miR-2898, pma-miR-138b, and novel-176) may play crucial roles in gonadal development in *T. blochii*. Our research identified a set of sex-biased miRNAs that might be regulatory factors affecting gonadal development in *T. blochii*. This provides new insight and may further our understanding the role of miRNAs in gonadal development of this economically valuable species.

**Abstract:**

The golden pompano (*Trachinotus blochii*) is a marine fish of considerable commercial importance in China. It shows notable sexual size dimorphism; the growth rate of females is faster than that of males. Therefore, sex-biased research is of great importance in *T. blochii* breeding. However, there have been few studies on sex differentiation and mechanisms underlying sex determination in *T. blochii*. MicroRNAs (miRNAs) play crucial roles in sex differentiation and determination in animals. However, limited miRNA data are available on fish. In this study, two small RNA libraries prepared from the gonads of *T. blochii* were constructed and sequenced. The RNA-seq analysis yielded 1366 known and 69 novel miRNAs with 289 significantly differentially expressed miRNAs (*p* < 0.05). Gene ontology (GO) analysis confirmed that the TFIIA transcription factor complex (GO: 0005672) was the most significantly enriched GO term. The Kyoto Encyclopedia of Genes and Genomes (KEGG) analysis showed that the differentially expressed miRNAs and target genes were mainly related to sex determination and gonadal developmental signaling pathways, specifically the Wnt signaling pathway, MAPK signaling pathway, and steroid biosynthetic pathway. MiRNA-mRNA co-expression network analysis strongly suggested a role for sex-biased miRNAs in sex determination/differentiation and gonadal development. For example, *gata4*, *foxo3*, *wt1*, and *sf1* genes were found to be regulated by bta-miR-2898; *esr2* and *foxo3* by novel_176, and *ar* by oar-let-7b. Quantitative real-time polymerase chain reaction analysis of selected mRNAs and miRNAs validated the integrated analysis. This study established a set of sex-biased miRNAs that are potential regulatory factors in gonadal development in *T. blochii*. These results provide new insight into the function of miRNAs in sex differentiation and determination in *T. blochii* and highlight some key miRNAs for future studies.

## 1. Introduction

MicroRNAs (miRNAs) are a class of small non-coding RNAs that have been considered to be crucial gene expression regulators in various species [1]. Mature miRNA sequences are approximately 20–24 nucleotides (nt) in length and they regulate gene expression by binding to the 3′-untranslated regions (3′-UTRs) of the target genes [2]. MiRNAs may have multiple target genes, and each target gene can be regulated by multiple miRNAs [3]. MiRNAs are essential to the regulation of biological processes including those involved in tissue development, apoptosis, and cell proliferation.

Sex differentiation and determination are crucial phylogenic processes in sexually reproducing animals, and they are regulated by many mechanisms [4]. In mammals, miRNAs are involved in reproduction, sex differentiation, gametogenesis, and embryogenesis [5,6,7,8,9]. Genes, social interaction, and environmental factors may all contribute to sex determination and differentiation in many fish species [10]. Teleosts, which are the largest group of vertebrates, have many diverse sexual systems [11]. Teleost miRNAs were first discovered in zebrafish [12], and several studies have reported the function of miRNAs in regulating gonadal development. Let-7 and miR-21 regulate the development of eggs in rainbow trout [13]. MiRNAs also participate in oocyte development, hydration, and competence, indicating their importance in the regulation of oogenesis [14]. MiR-141 and miR-429 have vital functions during testicular development and spermatogenesis in yellow catfish [15]. Let-7a, miR-143, and miR-202 are upregulated to induce differentiation of halibut testes [16]. Wang et al. suggested a possible role for miR-17-5p and miR-20a in estrogen production by preventing and enhancing the expression of *double-sex and mab-3-related transcription factor 1* (*dmrt1*) and the *cytochrome P450 family 19 subfamily A polypeptide 1a* (*cyp19a1a*), respectively [14]. MiR-138, miR-200a, and miR-338 may negatively regulate *cytochrome P450 family 17 subfamily A polypeptide 2* (*cyp17a2*), which is involved in the biosynthesis of *20β-dihydroxy-4-pregnen-3one* (*20β-P*), which plays a vital role in cell multiplication and spermatogenesis [14]. In this way, miRNAs may be novel regulators of sex differentiation and gonadal development.

*Trachinotus blochii* mainly grows in tropical and subtropical waters. After years of development, *T. blochii* has become one of the three major mariculture fish in China because of its fast growth, strong adaptability, good flavor, and high nutritional value [17]. There are no distinct morphological differences between female and male *T. blochii*, even at the mature stage [18]. This increases the difficulty of breeding parent fish and constructing its family. Therefore, it is important to know about the gonad development and reproductive regulation of *T. blochii*. A single nucleotide polymorphism (SNP) on chromosome 24 which is confirmed by whole-genome sequencing, is closely associated with phenotypic sex [19]. There have been few studies on the sex differentiation and determination of *T. blochii*, which has led to research into the sex-biased characteristics of *T. blochii* [20]. Many studies have verified that miRNAs are dimorphically expressed in fish [21]. However, the function of miRNAs in the reproductive development of *T. blochii* is still unknown. Clearly, establishing the role of miRNAs in gonadal development of *T. blochii* is significant.

In the present work, we performed RNA sequencing to identify specific miRNAs expressed in *T. blochii* gonads and investigated the differentially expressed miRNAs (DEMs) to identify miRNAs related to gonadal development and differentially expressed in the testes and ovaries. The present study may facilitate the analysis of the differences between males and females and reveal the molecular mechanisms underlying these differences. Females also grow faster than males and have higher nutritional value, and breeding female *T. blochii* has considerable economic value. The results may facilitate better comprehension of the functions of miRNAs in the process of *T. blochii* sex differentiation and gonadal development and provide basic data for the further study.

## 2. Materials and Methods

### 2.1. Experimental Animals and Sample Collection

The *T. blochii* samples were obtained from Blue Grain Technology Co., Ltd. (Sanya, Hainan Province, China). The fish were spawned in the same sea cage and were 10 months old at the time of the study. Our previous research (unpublished) showed ovary development to take place during the middle and late periods of stage Ⅱ and the testis during the end of stage Ⅲ. The average weight of the male *T. blochii* was 893.2 ± 43.8 g, and that of females was 1028.5 ± 51.2 g. We gathered gonadal tissues from nine females (three samples were placed in one microcentrifuge tube) and nine males (three samples per microcentrifuge tube), which were snap-frozen in liquid nitrogen, and stored at −80 °C until processing. All experimental procedures were conducted in accordance with the Guidelines for the Care and Use of Laboratory Animals in China. The Animal Experimentation Ethics Committee of Hainan University approved this protocol (protocol code HNUAUCC-2021-00007, 26 February 2021).

### 2.2. RNA Extraction, miRNA Library Construction and Sequencing

The gonad RNA was isolated by using the TRIZOL reagent (Invitrogen, Carlsbad, CA, USA), and DNA was eliminated using DNase I (New England Biolabs, New Ipswich, MA, USA). We used a NanoPhotometer^®^ spectrophotometer to detect RNA concentration and purity. We used 1% agarose gel electrophoresis to determine if the RNA was contaminated or degraded. After the samples were qualified, they were used to construct the sequencing libraries. Sequencing libraries were produced utilizing NEBNext^®^ Multiplex Small RNA Library Prep Set for Illumina^®^ (New England Biolabs) according to the recommendations of manufacturer, and index codes were appended to the affiliated sequences for each sample. Purified RNA (2 µL) was concatenated with 3′ and 5′ accommodators (Illumina, San Diego, CA, USA) utilizing T4 ligase (New England Biolabs) (1 μL), and then reverse transcribed into first-strand cDNA. The polymerase chain reaction (PCR) was used to amplify primers complementary to the adaptor sequences. The PCR products were purified by polyacrylamide gel electrophoresis (100 V, 80 min). To obtain the final small RNA sequencing library, DNA fragments corresponding to 140–160 bp (small non-coding RNA plus the 3′ and 5′ adaptors) were restored and dissolved in 8 μL of elution buffer. Each library was loaded into the Illumina Hiseq (Illumina, San Diego, CA, USA) lane for single-end sequencing.

### 2.3. Identification of miRNA

The quality control and read statistics of the original sequence were determined using FastQC [22]. Sequences with poor quality, no 3′ adaptor, 5′ adaptor contaminants, or trimmed sequences with short lengths (less than 18 nt) were here considered unordered reads and were eliminated before analysis. The clean reads Q20 (Q20 represent the percentage of bases with a Phred value greater than 20 in the total bases) and Q30 (Q30 represent the percentage of bases with a Phred value greater than 30 in the total bases) (Phred = −10log_10_(e)) and GC contents were computed, and high-quality, clean reads were applied to downstream analyses. These clean reads were mapped to the reference genome of *T. blochii* utilizing HISAT2 software. The reference genome of *T. blochii* was constructed in our laboratory and the data have not been published. Then, transfer RNAs, ribosomal RNAs, small nucleolar RNAs, and small nuclear RNAs were filtered from the mapped sequences through the Rfam (http://rfam.xfam.org/ (accessed on 27 October 2022)) and GenBank (http://www.ncbi.nlm.nih.gov (accessed on 27 October 2022)) databases. The residual sequences were estimated as conservative miRNAs referring to the miRBase database. The ratio of total rRNA was used as an indicator of sample quality. The rate of detection of high-quality plant samples was usually below 60%, and the rate of detection of animal samples was 40% [23]. To identify novel miRNAs, we submitted the ultimately unannotated sRNA reads utilizing MIREAP software (https://sourceforge.net/projects/mireap/ (accessed on 27 October 2022)) to evaluate the secondary structure of Dicer cleavage sites and map the minimum free energy Dicer cleavage sites and unnamed small RNA marker genomes.

### 2.4. Identification of the Differentially Expressed miRNAs

The expression levels of the miRNAs in the testis and ovaries were compared by normalizing the frequency of miRNA counts to transcripts per million (TPM) [24]. The differential expression analysis of two groups was accomplished utilizing the DESeq R package (3.0.3). The *p*-values were corrected using the method described by Benjamini and Hochberg to control the error rate. Genes with |log2 (fold change)|> 0 and padj < 0.05 between the two groups were considered DEMs.

### 2.5. Prediction and Functional Annotation of the DEM Target Genes

MiRanda with default settings was used to identify DEM target genes [25]. We combined the miRNA-seq in this study with that from our previous study and so finally identified miRNAs and target genes with negative regulatory relationships [20]. We used the Gene Ontology (GO) database (http://www.geneontology.org/ (accessed on 27 October 2022)) to annotate the biological processes of the DEMs target genes [26] and the Kyoto Encyclopedia of Genes and Genomes (KEGG) database (http://www.genome.jp/kegg/ (accessed on 27 October 2022)) to confirm which biochemical and signal transduction pathways were significantly related to the DEM target genes [27]. KOBAS software [28] was used to statistically test the enrichment of the target gene candidates in the KEGG pathways. Terms with correct *p*-values < 0.05 were considered significantly enriched.

### 2.6. Co-Expression Network Analysis

To clearly establish the potential regulatory relationship between gender-related miRNA-mRNA gene pairs, the gender-related gene pairs were screened by predicting miRNA target genes based on the correlation between the DEMs and mRNA, and by analyzing a co-expression network.

### 2.7. Quantitative Real-Time PCR (qPCR) Analysis

To verify the miRNA-seq accuracy and the miRNA and mRNA relationship correctness, seven DEM target genes and 10 DEMs were chosen at random for the qPCR experiment. The RNA of the qPCR samples was used to perform the miRNA sequencing. The primers for the target genes and the DEMs which were used are presented in Appendix A; β-actin served as the internal reference for the mRNAs and U6 as the internal reference for the miRNAs. qPCR was conducted on a LightCycler^®^ 480 II Instrument (Roche, Basel, Switzerland). The 20 μL reaction volume for mRNA quantification contained 10 μL of SYBR Green Master Mix (2×) (Q711, Vazyme, Nanjing, China), 0.4 μL of each sense and antisense primer (10 μM), 1 μL of cDNA (100 ng/µL), and 8.2 μL of ddH_2_O. The PCR amplification procedure was performed at 95 °C for 30 s, followed by 40 cycles at 95 °C for 5 s and 55 °C for 30 s. The quantification of miRNA for the reaction system was the same as for mRNA. The cycling parameters were 95 °C for 5 min, followed by 40 cycles at 95 °C for 10 s and 60 °C for 30 s. We estimated primer specificity according to melting curves and calculated relative expression levels of mRNA and miRNA using the 2^−∆∆Ct^ method [29].

### 2.8. Statistical Analysis

Data are rendered as mean ± standard deviation for three independent experiments. The independent sample *t*-test was used to detect differences using SPSS 22.0 software (SPSS Inc., Chicago, IL, USA). *p*-value < 0.05 was considered significant.

## 3. Results

### 3.1. miRNA Sequencing

To determine the expression profiles of miRNA in the *T. blochii* gonads, six total RNA libraries concerning the testis and ovaries were established and subjected to Illumina deep sequencing. In total, 88,466,133 raw reads were produced: 15,122,647, 15,619,201, and 19,226,171 reads from female libraries O1, O2, and O3, respectively, and 10,025,440, 14,336,950, and 14,135,724 reads from male libraries T1, T2, and T3, respectively. After quality filtering, 84,769,441 clean reads were obtained. The clean reads were mapped to the reference genome of *T. blochii* to produce the genomic distribution of the miRNAs. More than 72% of the clean reads were mapped to the reference genome of *T. blochii*. About 30% and 50% of the reads were mapped to the negative and positive strands of genome, respectively (Appendix A). The length distributions of the miRNAs in each sample are given in Figure 1. The miRNA read length distributions differed between the testis and ovary. The lengths of the miRNAs varied more in the ovary than in the testis. The most common length in the ovary was 27 nt, compared to 21 nt in the testis.

### 3.2. Micro RNA Identification and Novel miRNA Prediction

In total, 1366 known miRNAs were acquired from the gonads, including 437 3p-miRNA sequences and 457 5p-miRNA sequences. According to the length distributions of the known miRNAs, we learned that 22 nt miRNAs were the most abundant, followed by 21 nt miRNAs (Appendix A). We also obtained 69 novel miRNAs, and the length distribution of novel miRNAs was 22–23 nt (Appendix A). The nucleotide nt preference distributions in the novel miRNAs revealed that among the four bases, uracil (U) had the largest ratio, followed by adenosine (A). Family analysis for known and novel miRNAs identified 171 miRNA families comprising 1 to 110 members.

### 3.3. Differentially Expressed miRNAs

Development-related miRNA expression was determined for the six testis and ovary samples; 1435 miRNAs were obtained, including novel and known miRNAs. Overall, 289 miRNAs were predicated to be significantly differentially expressed between the ovary and testis groups in the gene expression differential display. The hcluster heatmap of the DEM levels is shown in Figure 2A, including 268 known and 21 novel miRNAs (Appendix A). Among them, 135 were downregulated and 154 were upregulated (Figure 2B).

### 3.4. Prediction of Differentially Expressed miRNA Target Genes

Based on the miRNA-seq in this study and mRNA-seq in our previous study [19], we obtained DEMs and differentially expressed mRNAs (DEGs) expressed in opposite directions. We combined this with target prediction and identified miRNA and target genes with negative regulatory relationships. We compared the interaction network of miRNAs to their target genes in the ovary group and the testis group, and 289 DEMs and 4062 DEGs formed 13,403 negatively correlated miRNA-mRNA pairs (Appendix A). Due to the negative regulatory mechanisms of miRNAs and their target genes, we identified a large number of miRNAs targeting sex-biased genes. After the analysis, the miRNAs were found to target one or more sex-biased genes, including pma-miR-138b (*piwi-like RNA-mediated gene silencing 1, piwil1*), oar-let-7b (*androgen receptor*, *ar*), novel_176 (*estrogen receptor 2*, *esr2* and *forkhead box O3*, *foxo3*), and bta-miR-2898 (*GATA binding protein 4*, *gata4*; *WT1 transcription factor*, *wt1*; *foxo3*; and *steroidogenic factor 1*, *sf1*). These four key sex-biased miRNAs and their target genes were randomly selected to verify that the targeted regulation was correct, and the qPCR validation results are given in Figure 3.

### 3.5. GO and KEGG Enrichment Analyses

GO and KEGG enrichment analyses were executed on the target genes of the DEMs according to the corresponding relationships between the miRNAs and their target genes. All DEGs annotated for GO analysis were divided into three categories of molecular functions, biological processes, and cellular components. GO term analysis showed that 117 GO terms were enriched (*p* < 0.05). Among biological processes, the putative target genes were significantly enriched in “transcription initiation from the RNA polymerase II promoter” (GO: 0006367). Among cellular components, the putative target genes were significantly enriched in “transcription factor TFIIA complex” (GO: 0005672). At the molecular function level, “enzyme regulator activity” (GO: 0030234) had more counts in target genes (223 genes) (Figure 4 and Appendix A).

A total of 162 KEGG pathways were detected in the KEGG enrichment analysis (Appendix A), and the top 20 pathways are presented in Figure 5A. The pathways consisted of those involved in gonadal and sexual development, consisting of the MAPK signaling pathway [30], the Wnt signaling pathway [31], and the steroid biosynthetic pathway [32].

To learn more about the functions of the DEMs, 20 miRNAs highly expressed in the gonads were selected to predict the enriched pathways. Most genes in the ovary were enriched in the MAPK [30] and Wnt signaling pathways [31] (Figure 5B and Appendix A). The sex differentiation and gonadal development pathways were also identified in the testis, including the transforming growth factor (TGF)-beta [33] and the peroxisome proliferator-activated receptor (PPAR) signaling pathways [34] (Figure 5C and Appendix A).

### 3.6. miRNA-mRNA Co-Expression Network Analysis

DEM target genes have been found to be mainly enriched in pathways related to sex differentiation and gonadal development signals, including the MAPK signaling pathway [30], Wnt signaling pathways [31], steroid biosynthesis pathway [32], and TGF-beta signaling pathway [33]. To further understand the role of the DEMs during gonadal development, co-expression network analysis using Cytoscape software was performed with the DEMs and mRNAs enriched in the sex differentiation and gonadal development pathways. The results shown in Appendix A reflect the four types of networks found: one mRNA associated with multiple miRNAs, one mRNA associated with one miRNA, multiple mRNAs associated with multiple miRNAs, and multiple mRNAs associated with one miRNA. The miRNA-mRNA co-expression relationship is shown in Figure 6.

### 3.7. Quantitative Real-Time PCR (qPCR) Validation

To validate the results of miRNA-seq, we performed a qPCR experiment to investigate the relative expression levels of nine randomly selected miRNAs (seven upregulated and two downregulated) from the testis and ovary groups. The tendencies in the expression of the nine miRNAs detected by qPCR were closely consistent with the results of miRNA-seq, which strengthened the results of this study (Figure 7).

## 4. Discussion

Micro RNAs regulate many physiological functions, such as cell metabolism, gonadal development, differentiation, and sex reversal. Gene expression at the post-transcriptional level can be regulated by miRNAs by binding to the target gene 3′ UTR region. They play a crucial role in the regulation of gonadal development [21]. In the present research, 1435 miRNAs were obtained and 289 significant DEMs were identified based on the RNA-seq results, including 268 known and 21 novel miRNAs. We identified two peaks in miRNA length distribution in our study. The length of the highest peak was 27–28 nt, and the length of the second-highest peak was 21–22 nt. This indicates a duality of miRNA length distribution in the gonad, which was consistent with the findings of previous studies on yellow catfish [35]. The main peak, with 27–28 nt reads in the gonad, was mainly attributable to the abundant expression of Piwi-interacting RNAs (piRNAs) [36], which are related to gene silencing, specifically transposon silencing [37]. In mammals, piRNAs play critical roles in germ cell development [38], and previous studies have found that piRNAs exist in the testes during both sexually immature and mature stages [39].

### 4.1. Known Sex-Biased miRNAs

Our results indicated that let-7b-5p and miR-2898 were more abundantly expressed in the ovaries of *T. blochii.* Let-7 is a regulator that may play a part in concerting mRNA stability and translation in bovine oocytes [39]. MiR-let-7 acts as a primary regulator in *Drosophila* sex determination and gonadal differentiation, maintaining ovarian function [40]. Studies on *Trachinotus ovatus* suggest that dre-let-7c-5p regulates ovarian development [41]. Let-7 is also highly expressed in the ovaries of *Smphysodon aequifasciatus* [42]. Our study suggested that let-7b-5p was highly expressed in the ovaries, indicating that it may take part in ovarian development in *T. blochii*. Studies have shown that *sf1* plays crucial roles in slider turtle sex differentiation and determination [43] and is an essential regulator of gonad development in Nile tilapia [44]. *Foxo3* has been shown to be involved in ovarian development in the *Gobiocypris rarus* [45], and it upregulates *cyp19a1a* during vitellogenesis in the orange-spotted grouper [46]. *Gata4* is not only required for gonad differentiation in tilapia but also important to gonad development and maturation [47]. It also performs a crucial function in sex differentiation in *Cynoglossus semilaevis* [48]. Our findings suggest that the female sex-biased genes *sf1*, *foxo3*, and *gata4* are regulated by bta-miR-2898, which has also been shown to be involved in the formation of the ovarian corpus luteum in cows [49]. This indicates that bta-miR-2898 may play a part in female gonad development in *T. blochii.* KEGG enrichment analysis has been performed on target genes of highly expressed miRNAs within the gonads. Most genes are enriched in the MAPK signaling pathway [30] and the Wnt signaling pathway [31] in the ovary. The target gene regulatory pathways of highly expressed miRNAs in the testis are mainly the TGF-beta signaling pathway [33] and the PPAR signaling pathway [34]. The Wnt signaling pathway is critical to mammalian female development [50]. The TGF-beta signaling pathway is activated in mammalian testis and plays a key role in promoting spermatogenesis and maintaining testicular differentiation [33]. These results suggest that related miRNAs may play an important role in gonadal development and differentiation.

We also found miR-130b-5p and miR-22 to be richly expressed in *T. blochii* ovaries, and miR-202, miR-145, miR-143, and miR-103b were more abundantly expressed in the testes. These miRNAs are also crucial to gonadal development in other species. MiR-22a-3p is involved in vitellogenesis and ovarian development in *Danio rerio* [51]. In addition, human studies have shown that miR-130b is benign in normal ovarian tissues, but miR-130b expression trends downward in malignant ovarian tumors [52]. This research suggest that miR-130b may play a key role in maintaining ovarian function in *T. blochii*. MiR-202 is upregulated to induce testis differentiation in zebrafish [53]. MiR-202 may also be involved in testicular differentiation in *T. blochii.* MiR-143, miR-145, and miR-202-3p have been detected during testis development and spermatogenesis in *Atlantic halibut* [54]. They are also the main miRNAs detected in the testis of *Nile tilapia* [10]. MiR-145 plays an important role in male differentiation by targeting male *SRY-box transcription factor 9* (*sox*9), which acts on Sertoli cells [55]. MiR-103 is expressed in both XY and YY testis of yellow catfish [56]. MiR-103 has been found to be expressed at high levels in the testis of *Atlantic halibut* [54]. These results suggest that these miRNAs may also play a crucial role in the gonadal development of *T. blochii*. The functions of these miRNAs in *T. blochii* need to be verified through further investigation.

### 4.2. Novel Sex-Biased miRNAs

We found novel miRNAs, such as novel_176, novel_133, novel_145, and others. Novel_176 was found to be expressed at higher levels in the ovary than in the testis, and it is a potential novel miRNA in gonad of *T. blochii*. KEGG analysis confirmed that novel_176 and target genes are related to sex determination and gonadal development signaling pathways, such as oocyte meiosis (sdu04114), the TGF-beta signaling pathway (sdu04350), and the FoxO signaling pathway (sdu04068). In our study, we found novel-176 regulated *esr2* and *foxo3*. *Foxo3* is in the FoxO signaling pathway. Studies have shown that the FoxO signaling pathway, which contains a transcription factor family responsible for suppressing the expression of genes related to cell growth, proliferation, and differentiation, is upregulated in female hybrid tilapia [57]. Novel-176 may be related to female sex differentiation. *Esr2* is a sex-determining gene in Carangidae fishes. It is highly expressed in the testis. Estrogen receptor expression is male-biased in the bluehead wrasse [58], *Nile tilapia* [59], and rainbow trout [60], suggesting that novel-176 may also be related to gonadal development in males. Novel_133 targets *sox9*. *Sox9* is a crucial gene in male sex differentiation and gonadal development. Studies have shown that the *sox9* gene is expressed in the testis of mature individuals and plays an important role in spermatogenesis [61]. The *sox9* gene was detected in the gonads of *Pelteobagrus fulvidraco*, carp, and *Monopterus albus* [62]. This suggests that novel_133, which regulates *sox9*, may play a crucial role in the differentiation and development of male gonads. Novel_145 regulates the expression of *wt1*, which has been found to have high expression in developing and adult testis of catfish and played an important role in spermatogenesis [63]. In addition, *wt1* has been found to be a testis-biased gene in *Silurus asotus* [64], *Cyprinus carpio*, and tilapia [65,66]. In our study, novel_145 was also highly expressed in the testis of *T. blochii*, indicating that novel_145 may play an essential role in the development of testis. In conclusion, these novel miRNAs can be the significant candidates for further studies of gonadal development in *T. blochii.*

## 5. Conclusions

Transcriptome sequencing was performed on the testes and ovaries of *T. blochii* in the present work. RNA-seq analysis yielded 1366 known and 69 novel miRNAs with 289 significant DEMs (*p* < 0.05). GO analysis confirmed that the transcription factor TFIIA complex (GO: 0005672) was the most significantly enriched GO term. KEGG analysis showed that the target genes of the DEMs are mainly enriched in pathways related to sex differentiation and gonadal development signals, consisting of the MAPK signaling pathway, Wnt signaling pathways, and steroid biosynthetic pathways. The co-expression network analysis of miRNA-mRNA indicated that some miRNAs (oar-let-7b, bta-miR-2898, pma-miR-138b, and novel-176) may play key roles in *T. blochii* gonadal development. These results will improve our understanding of the molecular mechanism of gonadal development and facilitate genetic studies and breeding of *T. blochii*.

## Figures and Tables

**Figure 1 animals-12-03342-f001:**
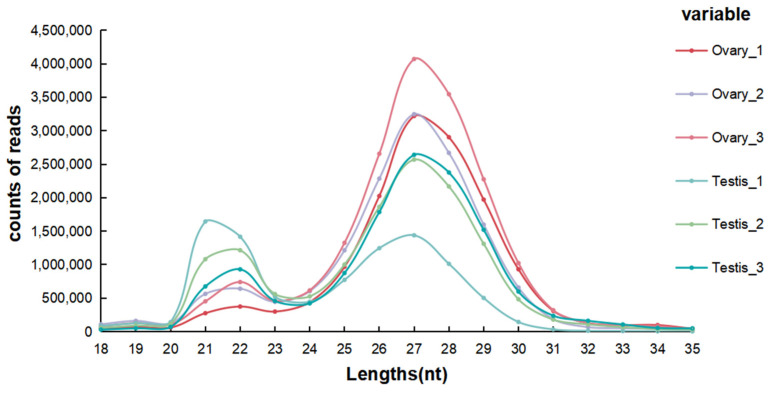
Read length distributions of the miRNAs in the six libraries.

**Figure 2 animals-12-03342-f002:**
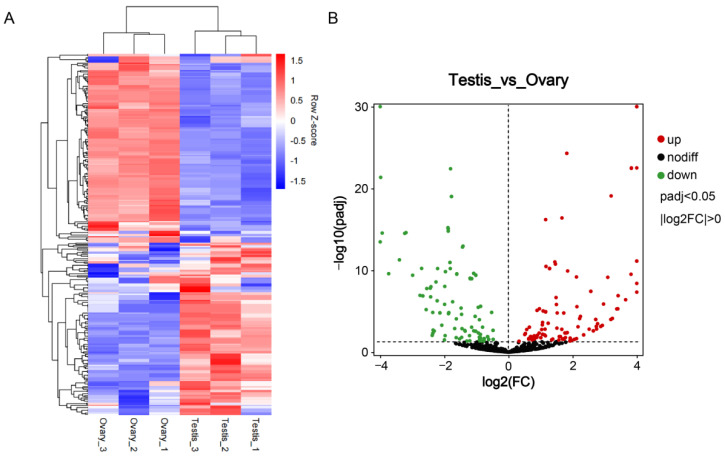
(**A**): Heatmap of all miRNA (DEMs) differentially expressed between the ovary and testis groups. Red represents highly expressed miRNAs and blue represents minimally expressed miRNAs. (**B**): Volcano plot of the miRNAs in the ovary and testis groups.

**Figure 3 animals-12-03342-f003:**
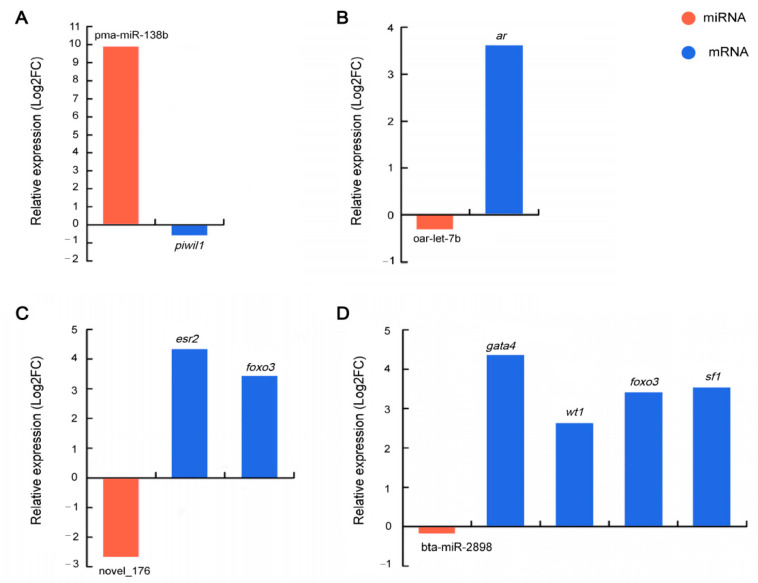
Four key sex-biased miRNAs and their target genes verified by qPCR. (**A**): pma-miR-138b regulated *piwil1*. (**B**): oar-let-7b regulated *ar*. (**C**): novel_176 regulated *esr2* and *foxo3*. (**D**): bta-miR-2898 regulated *gata4*, *wt1*, *foxo3*, and *sf1*.

**Figure 4 animals-12-03342-f004:**
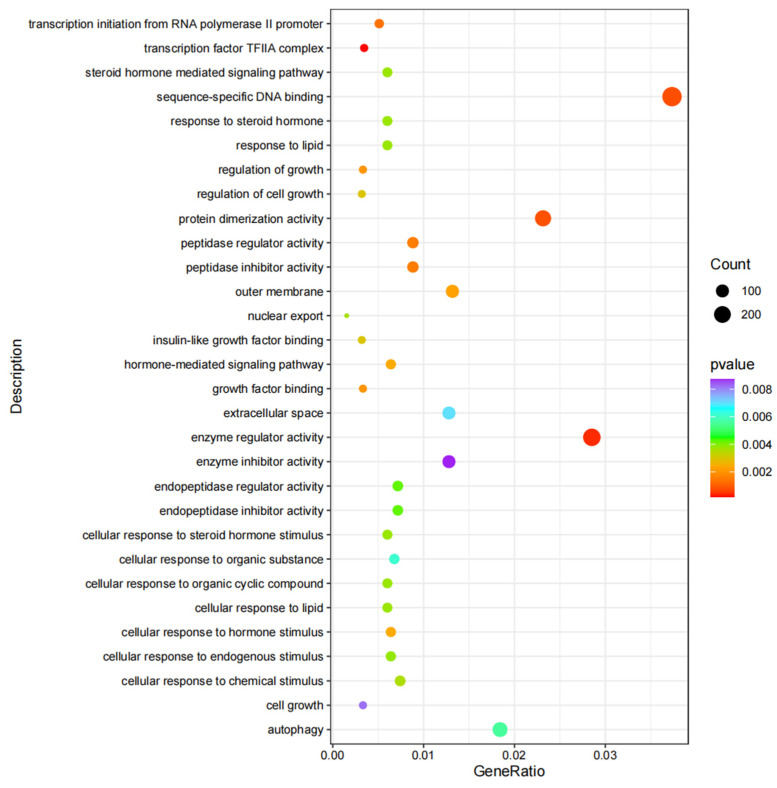
GO enrichment analyses of the differentially expressed miRNA target genes.

**Figure 5 animals-12-03342-f005:**
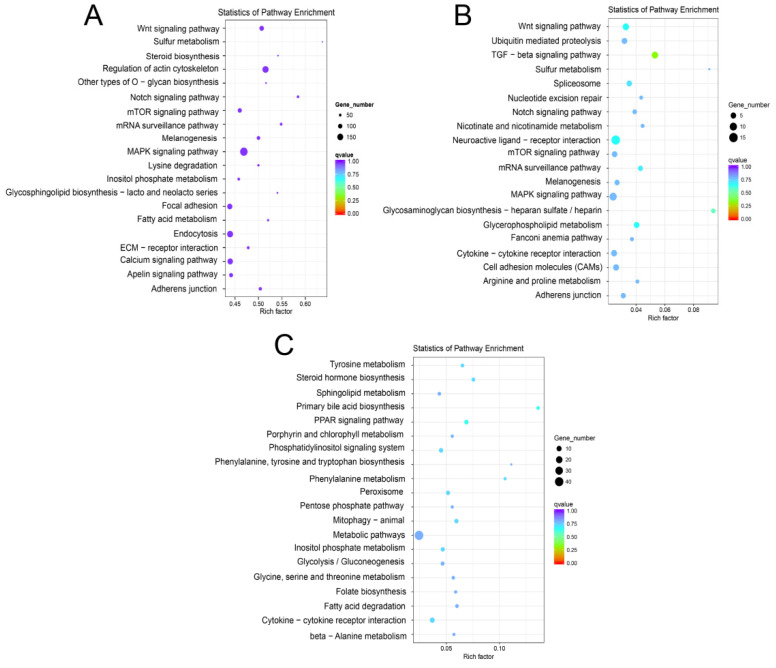
KEGG enrichment results. (**A**): Differentially expressed miRNA target genes. (**B**): The target genes of the highly expressed miRNAs in the ovaries. (**C**): The target genes of the highly expressed miRNAs in the testes.

**Figure 6 animals-12-03342-f006:**
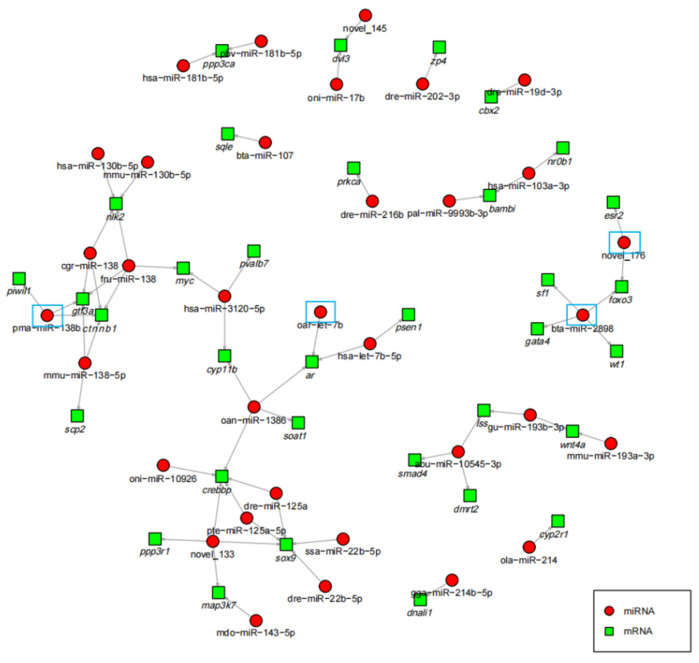
Differentially expressed miRNA–mRNA negative correlation network. The blue boxes are the four relatively important miRNAs, red circle represents miRNA, and green box represents target gene.

**Figure 7 animals-12-03342-f007:**
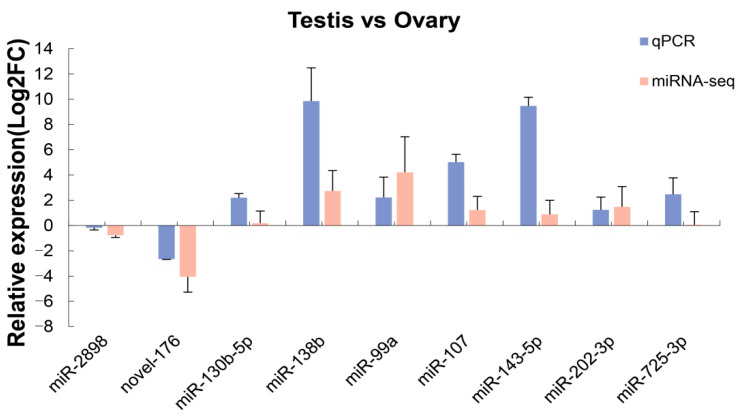
Comparison of the expression levels of nine DEMs by miRNA-seq and quantitative real-time PCR.

## Data Availability

The data used and/or analyzed during the current study have submitted to NCBI, the acception number were: SAMN30674718, SAMN30674719, SAMN30674720, SAMN30674721, SAMN30674722, SAMN30674723.

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
