# Peer review of "Identification and Characterization of Sex-Biased miRNAs in the Golden Pompano (Trachinotus blochii)"

_animals, 2022, doi:10.3390/ani12233342_

Round 1

Reviewer 1 Report (New Reviewer)

In this manuscript, the authors compared miRNA expression levels in the testes and ovaries of T. blochii to investigate sex differentiation and determination. They found several miRNAs that may act as regulatory factors affecting gonadal development in T. blochii.

LINE 25-26

In the Abstract, it is stated that the growth rate is faster in females, but this was not included in the text. It would be better to explain in the discussion section how the results obtained in this study affect this difference in growth rates.

Fig. 1

The colors of the lines in the graphs for Ovary and Testis are similar, making it difficult to tell the difference between them.

LINE202

Manuscript states that 21 nt is most common in testis, but 27 nt appears to be more common in all samples in testis.

Fig. 2B

The legend says "|log2FC| > 0" which should be "|log2FC| > 1". The vertical dotted lines in the graph are at 1 and -1.

Fig. 6

The "blue box" in the legend is not found in the figure.

LINE 132   The manuscript states "trimmed sequences with short length", but what is the exact length?

LINE 133   What do Q20 and Q30 indicate? Is it the number of bases with quality values greater than 20 and 30?

LINE 135             From which database was the reference genomic of T. blochii obtained?

LINE 220             “and154”   should be “and 154”

Author Response

Thank you so much for the reviewer's valuable comments. We have revised our manuscript according to all the comments and suggestions. The following are our comments and responses:

Reviewer #1:

LINE 25-26

In the Abstract, it is stated that the growth rate is faster in females, but this was not included in the text. It would be better to explain in the discussion section how the results obtained in this study affect this difference in growth rates.

Answer: Thanks for your suggestion. In this study, we performed miRNA-seq on the gonads of T. blochii and obtained DEMs and we focused on sex-biased miRNA. We observed the phenomenon of faster growth of females than males in T. blochii and the results have been published in another article (Sun et al., 2022).

Reference:

Sun, J.L.; Song, F.B.; Wang, L.; Zhang, W.W.; Chen, Y.M.; Zhou, L.; Gui, J.F.; Luo, J. Sexual size dimorphism in golden pompano (Trachinotus blochii): Potential roles of changes in energy allocation and differences in muscle metabolism. Front. Mar. Sci. 2022, 9, 1-18.

Fig. 1

The colors of the lines in the graphs for Ovary and Testis are similar, making it difficult to tell the difference between them.

Answer: Thanks for your suggestion. We have adjusted the colors of the lines in Fig. 1.

LINE202

Manuscript states that 21 nt is most common in testis, but 27 nt appears to be more common in all samples in testis.

Answer: Thank you. Mature miRNA sequences are approximately 20-24 nucleotides (nt) in length. As mentioned in discussion, the main peak, with 27-28 nt reads, was mainly attributable to the abundant expression of Piwi-interacting RNAs (piRNAs). Therefore, we stated that 21 nt is most common in testis in manuscript.

Fig. 2B

The legend says "|log2FC| > 0" which should be "|log2FC| > 1". The vertical dotted lines in the graph are at 1 and -1.

Answer: Thank you for pointing out this issue. We use | log2FC | > 0 as standard to analyze the results. When we use the standard “|log2FC| > 1”, The result data we obtained was too little, so the analytical standard was changed. The vertical dotted lines in the graph are at 1 and -1 in the graph was due to the error in the standard setting during the mapping process. We sincerely apologize for the error and have revised it in manuscript.

Fig. 6

The "blue box" in the legend is not found in the figure.

Answer: Thank you for pointing out this issue. We have revised it in manuscript.

LINE 132 The manuscript states "trimmed sequences with short length", but what is the exact length?

Answer: Thank you. The exact length is less than 18 nt and we have added it in manuscript.

LINE 133 What do Q20 and Q30 indicate? Is it the number of bases with quality values greater than 20 and 30?

Answer: Thank you.

Q20 represent the percentage of bases with a Phred value greater than 20 in the total bases (Phred = -10log10(e)).

Q30 represent the percentage of bases with a Phred value greater than 30 in the total bases (Phred = -10log10(e)).

LINE 135 From which database was obtained?

Answer: Thank you. The reference genome obtained from genome sequencing of T. blochii in our laboratory and the data have not been published.

LINE 220 “and154” should be “and 154”

Answer: Thank you for pointing out this issue. We have revised it in manuscript.

Reviewer 2 Report (Previous Reviewer 2)

The authors have made all the necessary changes to their manuscript, and I have no further comments. The manuscript can be now published as is.

Author Response

Thank you so much for the reviewer's valuable comments. We have revised our manuscript according to all the comments and suggestions. The following are our comments and responses:

Reviewer #2:

The authors have made all the necessary changes to their manuscript, and I have no further comments. The manuscript can be now published as is.

Answer: Thank you so much for your valuable comments and all your efforts on our manuscript.

Round 2

Reviewer 1 Report (New Reviewer)

LINE133

The author needs to explain in this section what Q20 and Q30 indicate

LINE135

The authors should explain in this section that the reference genome of T. blochii was constructed in their lab.

Author Response

Thank you so much for the reviewer's valuable comments. We have revised our manuscript according to all the comments and suggestions. The following are our comments and responses:

Reviewer #1:

LINE133

The author needs to explain in this section what Q20 and Q30 indicate

Answer: Thanks for your suggestion. We have revised it in manuscript.

LINE135

The authors should explain in this section that the reference genome of T. blochii was constructed in their lab.

Answer: Thanks for your suggestion. We have revised it in manuscript.

This manuscript is a resubmission of an earlier submission. The following is a list of the peer review reports and author responses from that submission.

Round 1

Reviewer 1 Report

Shi et al. conducted the RNA-seq of ovaries and testis of golden pompano to identify both known and novel miRNAs, and focused on sex-specific expressed miRNAs to understand the role of miRNA in gonadal development and sex determination. Some key sex-biased miRNAs and their target genes were identified. Some results are interesting and novel.However, some statements of detailed analyses and figure description could be improved to make all manuscript more sense. 

Major comments/questions:

1.     In section 2.1, authors mentioned the average weight of male and female samples, where the females are heavier/larger than males. However, the authors also mentioned that the importance of this study is that there is no distinct morphological difference between female and male golden pompano. This seems conflict to me. Have you done the statistical analysis to see if there is different size or weight between males and females. If there is, how authors could explain. Another problem will be more severe: If the detected DEM is because of size or weight instead of sex?

2.     Was the RNA-seq strand-specific? If yes, please clarify in Method section.

3.     Why the raw reads from male library were significantly less than female, especially T1?

4.     What is the red-blue scale in Fig. 2? This should be marked clearly.

5.     In Volcano plot (Fig .3), it will be much better to mark the dot with low p-value and high log2FoldChange.

6.     Author descripted prediction of target genes of miRNAs were conducted by “miRanda with default settings” in Method section. However, this prediction step in section 3.4 seems different (line 223-225). This is one of the most important analyses in this study but I am confused which one is right and what the authors virtually did!

7.     Line 230-235 and Fig 4, why these four miRNAs were keys? How did you know the genes are the target? Why did you select these four for detailed analysis?

8.     Fig.9, why some miRNAs were enclosed? No explanation in figure legend.

9.     Fig. 10, I understand the goal. But what is the positive log2FC means? Which is the baseline? Male or female? Normalization should be done, where set control as one. Also, I think there were triplicates of the samples, why not error bar here? Too many details are lacking.

Minor comments:

1.     Line 64: Seems lack of period between two sentences.

2.     Line 88: two periods, need to delete one.

3.     Authors should check the typos and punctuation carefully.

Author Response

Thank you so much for the reviewer's valuable comments. We have revised our manuscript according to all the comments and suggestions. The following are our comments and responses and we have also provided the responses in the attachment.

Reviewer #1:

Major comments/questions:

  1. In section 2.1, authors mentioned the average weight of male and female samples, where the females are heavier/larger than males. However, the authors also mentioned that the importance of this study is that there is no distinct morphological difference between female and male golden pompano. This seems conflict to me. Have you done the statistical analysis to see if there is different size or weight between males and females. If there is, how authors could explain. Another problem will be more severe: If the detected DEM is because of size or weight instead of sex?

Answer: Thank you for pointing out this issue. In this study, the female T. blochii was 1,028.5 ± 51.2 g and that of males was 893.2 ± 43.8 g, a significant difference between the growth of males and females was observed. However, we can’t determine the sex of T. blochii based on their size accurately, so there is no distinct morphological difference between males and females. And the research have been published (Sun et al., 2022).

In addition, as for the detected DEM may be due to size or weight, thank you again for your suggestion. In this study, we performed miRNA-seq on the gonads of T. blochii and obtained DEMs. DEMs were mainly sex-biased, of course, there may be some weight-related miRNAs, but we focused on sex-biased miRNA. Therefore, combined with the existing results of sex-biased miRNAs and target genes, we finally selected sex-related miRNAs and target genes to construct co-expression network.

Reference:

Sun, J.L.; Song, F.B.; Wang, L.; Zhang, W.W.; Chen, Y.M.; Zhou, L.; Gui, J.F.; Luo, J. Sexual size dimorphism in golden pompano (Trachinotus blochii): Potential roles of changes in energy allocation and differences in muscle metabolism. Front. Mar. Sci. 2022, 9, 1-18.

  1. Was the RNA-seq strand-specific? If yes, please clarify in Method section.

Answer: Thank you for pointing out this issue. The small RNA library is based on the special structure of the 3' and 5' ends of small RNA (complete phosphate group at the 5' end and hydroxyl group at the 3' end), we take total RNA as the starting sample, directly add connectors to both ends of small RNA, and then inversely transcribes to synthesize cDNA. Therefore, the RNA-seq does not have strand-specificity.

  1. Why the raw reads from male library were significantly less than female,especially T1?

Answer: Thank you. For the data results of miRNA, each sample order is 10M data volume, and no additional testing has been carried out. The difference in output is due to the different quality of the library. If the quality of the library is excellent, it will lead to more output. Thus, the reason for the lower raw reads of T1 may be the quality of the library was slightly poor, and from the correlation results of our sample, the correlation is high.

  1. What is the red-blue scale in Fig. 2? This should be marked clearly.

Answer: Thank you. Red represents the highly expressed miRNA and blue represents the low-expressed miRNA in Fig. 2, and we have revised as your suggestion.

  1. In Volcano plot (Fig .3), it will be much better to mark the dot with low p-value and high log2FoldChange.

Answer: Thanks for your suggestion, and we have marked the dot with low p-value and high log2FoldChange in Fig . 3

  1. Author descripted prediction of target genes of miRNAs were conducted by “miRanda with default settings” in Method section. However, this prediction step in section 3.4 seems different (line 223-225). This is one of the most important analyses in this study but I am confused which one is right and what the authors virtually did!

Answer: Thank you for pointing out this issue. Regarding target gene prediction, we proceed as follows: Firstly, we predicted the target genes of miRNA by miRanda. Based on the fact that miRNA is a post-transcriptional regulation of mRNA, it can inhibit or even silence the expression of its target genes, so the negative regulation relationship pairs were selected for analysis. Secondly, we obtained DEMs and DEGs with opposite expression directions based on miRNA-seq in this study and mRNA-seq in our previous study. Finally, combining the above two steps, we get miRNAs and target genes with negative regulatory relationships. Thank you again for pointing this issue and we have refined the prediction step in the manuscript.

  1. Line 230-235 and Fig. 4, why these four miRNAs were keys? How did you know the genes are the target? Why did you select these four for detailed analysis?

Answer: Thank you very much.

(1)Our goal is to identify sex-biased miRNAs, after target gene prediction and negative regulation analysis, we obtained miRNAs and target genes. Previous studies have indicated that the four target genes were sex-biased (Ramsey et al., 2007; Cao et al., 2022; Li et al., 2012; Liu et al., 2015; Yan et al., 2019; Baron et al., 2008; Shen et al., 2020; Anitha et al., 2019; Tao et al., 2018), so we presumed that miRNAs regulating target genes were sex-biased. 

(2)And target gene prediction was performed as follows: Firstly, we predicted the target genes of miRNA by miRanda. Based on the fact that miRNA is a post-transcriptional regulation of mRNA, it can inhibit or even silence the expression of its target genes, so the negative regulation relationship pairs were selected for analysis. Secondly, we obtained DEMs and DEGs with opposite expression directions based on miRNA-seq in this study and mRNA-seq in our previous study. Finally, combining the above two steps, we get miRNAs and target genes with negative regulatory relationships.

(3)We focused on sex-biased miRNAs and presumed that these four were sex-biased based on existing studies. Thus, we selected these four for detailed analysis.

Reference:

Ramsey, M.; Shoemaker, C.; Crews, D. Gonadal expression of sf1 and aromatase during sex determination in the red-eared slider turtle (Trachemys scripta), a reptile with temperature-dependent sex determination. Differentiation 2007, 75, 978–991.

Cao, Z.M.; Qiang, J.; Zhu, J.H.; Li, H.X.; Tao, Y.F.; He, J.; Xu, P.; Dong, Z.J. Transcriptional inhibition of steroidogenic factor 1 in vivo in Oreochromis niloticus increased weight and suppressed gonad development. Gene 2022, 809, 1-14.

Li, J.Z.; Chen, W.J.; Wang, D.S.; Zhou, L.Y.; Sakai, F.; Guan, G.J.; Nagahama, Y. Gata4 is involved in the gonadal development and maturation of the teleost fish tilapia, Oreochromis niloticus. J. Reprod. Dev. 2012, 58, 237-242.

Liu, H.; Lamm, M.S.; Rutherford, K.; Black, M.A.; Godwin, J.R.; Gemmell, N.J. Large-scale transcriptome sequencing reveals novel expression patterns for key sex-related genes in a sex-changing fish. Biol. Sex Differ. 2015, 6, 26.

Yan, L.X.; Feng, H.W.; Wang, F.L.; Lu, B.Y.; Liu, X.Y.; Sun, L.; Wang, D.S. Establishment of three estrogen receptors (esr1, esr2a, esr2b) knockout lines for functional study in Nile tilapia. J. Steroid Biochem. Mol. Biol. 2019, 191, 1-9.

Baron, D.; Houlgatte, R.; Fostier, A.; Guiguen, Y. Expression profiling of candidate genes during ovary-to-testis trans-differentiation in rainbow trout masculinized by androgens. Gen. Comp. Endocrinol. 2008, 156, 369-378.

Shen, F.F.; Long, Y.; Li, F.Y.; Ge, G.D.; Song, G.L.; Li, Q.; Qiao, Z.G.; Cui, Z.B. De novo transcriptome assembly and sex-biased gene expression in the gonads of Amur catfish (Silurus asotus). Genomics 2020, 112, 2603-2614.

Anitha, A.; Gupta, Y.R.; Deepa, S.; Ningappa, M.; Rajanna, K.B.; Senthilkumaran, B. Gonadal transcriptome analysis of the common carp (Cyprinus carpio): identifification of difffferentially expressed genes and SSRs. Gen. Comp. Endocrinol. 2019, 279, 67-77.

Tao, W.J.; Chen, J.L.; Tan, D.J.; Yang, J.; Sun, L.; Wei, J.; Conte, M.A.; Kocher, T.D.; Wang, D.S. Transcriptome display during tilapia sex determination and differentiation as revealed by RNA-Seq analysis. BMC Genomics 2018, 19, 1-12.

  1. Fig. 9, why some miRNAs were enclosed? No explanation in figure legend.

Answer: Thank you. Circled by the blue box are exactly the four relatively important miRNAs we mentioned in 3.4, red circle represents miRNA and green box represents target genes. And we have added explanation in figure legend.

  1. Fig. 10, I understand the goal. But what is the positive log2FC means? Which is the baseline? Male or female? Normalization should be done, where set control as one. Also, I think there were triplicates of the samples, why not error bar here? Too many details are lacking.

Answer: Thank you for pointing out this issue. Our standard is testis vs ovary and log2FC means that we take the values of the log2FC of qPCR and miRNA-seq for plotting. We have modified Fig. 10 and added the missing details.

Minor comments:

  1. Line 64: Seems lack of period between two sentences.

Answer: Thank you. We have revised according to your suggestion.

  1. Line 88: two periods, need to delete one.

Answer: Thank you. We have revised according to your suggestion.

  1. Authors should check the typos and punctuation carefully.

Answer: Thank you. We have checked the typos and punctuation carefully in the whole manuscript and corrected the mistakes.

Reviewer 2 Report

The comments are in the attachment.

Author Response

Thank you so much for the reviewer's valuable comments. We have revised our manuscript according to all the comments and suggestions. The following are our comments and responses and we have also provided the responses in the attachment.

Reviewer #2:

Major comments:

The study suffers from a similar problem that many of the publications regarding transcriptomics suffer from; it is written as if the whole point of the article was to do transcriptomics for the sake of doing transcriptomics, and not to actually answer any questions or advance the field in any way. The authors need to consider which questions is their study answering and why did they do it. Near the aims statement, they need to mention how can their study be utilized, i.e., what can it be used for. Also, in the Discussion section, the authors simply state that the miRNAs differentially expressed were similar to other studies. However, after doing GO and KEGG, they should mention which important biological pathways were affected, and what does that mean. Basically, the authors should pay attention to the fact of why did they do their study, and what does their study actually mean and contribute, rather than focusing on simply stating what they observed.

Answer: Thank you very much. In this study, we explored the differentially expressed miRNAs in the testes and ovaries at the transcriptome level, which is important for analyzing the differences between males and females and revealing the molecular mechanism of these differences. In addition, females grow faster and have high nutritional value, and breeding whole female T. blochii has higher economic value. Therefore, we conducted relevant research to provide basic data for the futher study. And we have revised it in the introduction and complemented the sex-related signaling pathway in the discussion section according to your suggestions.

Minor comments:

Line 67: Wang et al. [14] suggested …

Answer: Thank you. We have revised according to your suggestion.

Line 75: Write here the colloquial name and the full latin name of the species as it is its first mention in the text (apart from the abstract).

Answer: Thank you. We have revised according to your suggestion.

Line 75: … mainly grows …

Answer: Thank you. We have revised according to your suggestion.

Line 78: latin name in italics. Check this throughout the manuscript as this mistake was repeated several times in this paragraph.

Answer: Thank you. We have checked the whole manuscript and revised according to your suggestion.

Line 87-88: Please describe in more detail why is this study important. What will this clarification of miRNAs in this study mean? Could it be used as sex markers?

Answer: Thank you. In this study, we explored the differentially expressed miRNAs in the testes and ovaries at the transcriptome level, which is important for analyzing the differences between males and females and revealing the molecular mechanism of these differences. In addition, females grow faster and have high nutritional value, and breeding whole female T. blochii has higher economic value. Therefore, we conducted relevant research to provide basic data for the further study. And in T. blochii, miRNA can’t be used as sex markers.

Line 96: Please write this section in the past tense (several sentences are written in the present tense).

Answer: Thank you. We have revised according to your suggestion.

Line 97: Please state at which stage of development / sexual maturation were these individuals.

Answer: Thank you. According to the previous findings of our research group (the paper has not been published), the ovary development is in the middle and late period of stageⅡ, and the testis is in the end of stage Ⅲ of the individuals at this time. And we have added the stage of development of these individuals in manuscript.

Line 98: The samples were obtained from …

Answer: Thank you. We have rewritten the sentences as your suggestion.

Lines 105-106: It would be preferable to present the actual number of the licence.

Answer: Thank you. We have added the actual number of the license in the manuscript as your suggestion.

Line 111: … we used …

Answer: Thank you. We have rewritten the sentences as your suggestion.

Line 172: ddH2O

Answer: Thank you. We have revised according to your suggestion.

Line 216: can be   (as A and B).

Answer: Thank you. We have merged Fig. 2 and 3 into one.

Line 225: Where did the DEGs come from? In the M&M section only miRNA-Seq was described, not the regular mRNA-Seq. Please describe how did you get these results, and if they are from a previous study, please refer to it in the M&M section, and cite it in the Results section.

Answer: Thank you. DEGs are from the previous study, and we have rewritten the sentences as your suggestion and cited it in the results section.

Line 237: Please describe in the caption what is displayed in each of the panels

Answer: Thank you. We have revised according to your suggestion.

Line 257: Figures 6, 7 and 8 can be merged into one with as different panels.

Answer: Thank you. We have merged 6, 7 and 8 into one.

Line 270: Please describe in more detail to which major signaling pathways did these networks belong to, and what do .

Answer: Thank you. These networks belong to the MAPK signaling pathway, Wnt signaling pathways, the Steroid biosynthesis pathway, and TGF-beta signaling pathway and they mainly regulated sex differentiation and gonadal development. We have added them in manuscript.

Line 306: Please discuss if similar patterns of other studies were observed in this study as well. And whether any of the effects of other sex-biased mRNAs from other studies were observed in this one as well.

Answer: Thank you for pointing out this issue. We have compared our findings of sex-biased miRNAs and mRNAs with those of other studies in the manuscript, they have similar patterns (Xu., 2018; Gecaj et al., 2017; Yang., 2011; Zhang et al., 2017). And the functions of these miRNAs and target genes in T. blochii need to be further verified.

Reference:

Xu, Z. Expression analysis of mRNAs and miRNAs in gonads of Symphysodon aequifasciatus. Shanghai Ocean University, Shanghai, 2018.

Gecaj, R.M.; Schanzenbach, C.I.; Kirchner, B.; Pfaffl, M.W.; Riedmaier, I.; Cullen, R.Y.; Berisha, B. The dynamics of microRNA transcriptome in Bovine corpus luteum during its formation, function, and regression. Front. Genet. 2017, 8, 213-229.

Yang, C. Study on the mechanism of methylation regulation of miR-130b in chemotherapy resistance of ovarian cancer. Huazhong University of Science and Technology, Wuhan, 2011.

Zhang, J.; Liu, W.; Jin, Y.; Jia, P.; Jia, K.; Yi, M. MiR-202-5p is a novel germ plasm-specific microRNA in zebrafish. Sci. Rep. 2017, 7, 1-7.

Line 344: please delete ‘and so on’, and replace with ‘and others’ or similar.

Answer: Thank you. We have revised according to your suggestion.

Round 2

Reviewer 1 Report

Authors answered my questions and addressed problems

Author Response

Reviewer #1:

Comments and Suggestions for Authors

Authors answered my questions and addressed problems

Thank you so much for your valuable comments and all your efforts on our manuscript.

Reviewer 2 Report

The authors have adequately responded to all my comments, and the manuscript can be published after only few minor corrections:

Title: in the golden pompano

Line 91: two times word ‘help’

Line 230: two punctuations

Figures 5 and 6: The figures are of very poor quality. Higher resolution images need to be provided before publication! I would extend this recommendation to other Figures as well.

Line 300: Micro RNAs regulate …

Author Response

Reviewer #2:

Comments and Suggestions for Authors

The authors have adequately responded to all my comments, and the manuscript can be published after only few minor corrections:

Thank you so much for your valuable comments and all your efforts on our manuscript. We have revised our manuscript according to your comments and suggestions. The following are our comments and responses:

Title: in the golden pompano

Answer: Thank you. We have revised as your suggestion.

Line 91: two times word ‘help’

Answer: Thank you for pointing out this issue. I am so sorry to write it wrong and we have corrected it.

Line 230: two punctuations

Answer: Thank you for pointing out this issue and we have corrected it.

Figures 5 and 6: The figures are of very poor quality. Higher resolution images need to be provided before publication! I would extend this recommendation to other Figures as well.

Answer: Thank you for pointing out this issue. We have improved the clarity of the images with a resolution of 300 ppi and provided the original pictures in the form of an attachment.

Line 300: Micro RNAs regulate …

Answer: Thank you. We have revised as your suggestion.
